# ZERO-SHOT RECOMMENDER SYSTEMS

## ABSTRACT

Performance of recommender systems (RecSys) relies heavily on the amount of training data available. This poses a chicken-and-egg problem for early-stage products, whose amount of data, in turn, relies on the performance of their RecSys. In this paper, we explore the possibility of zero-shot learning in RecSys, to enable generalization from an old dataset to an entirely new dataset. We develop an algorithm, dubbed ZEro-Shot Recommenders (ZESREC), that is trained on an old dataset and generalize to a new one where there are *neither overlapping users nor overlapping items*, a setting that contrasts typical cross-domain RecSys that has either overlapping users or items. Different from previous methods that use categorical item indices (i.e., item ID), ZESREC uses items' generic features, such as natural-language descriptions, product images, and videos, as their continuous indices, and therefore naturally generalizes to any unseen items. In terms of users, ZESREC builds upon recent advances on sequential RecSys to represent users using their interactions with items, thereby generalizing to unseen users as well. We study three pairs of real-world RecSys datasets and demonstrate that ZESREC can successfully enable recommendations in such a zero-shot setting, opening up new opportunities for resolving the chicken-and-egg problem for data-scarce startups or early-stage products.

## 1 INTRODUCTION

Many large scale e-commerce platforms (such as Etsy, Overstock, etc.) and online content platforms (such as Spotify, Overstock, Disney+, Netflix, etc) have such a large inventory of items that showcasing all of them in front of their users is simply not practical. In particular, in the online content category of businesses, it is often seen that users of their service do not have a crisp intent in mind unlike in the retail shopping experience where the users often have a clear intent of purchasing something. The need for personalized recommendations therefore arises from the fact that not only it is impractical to show all the items in the catalogue but often times users of such services need help discovering the next best thing — be it the new and exciting movie or be it a new music album or even a piece of merchandise that they may want to consider for future buying if not immediately.

Modern personalized recommendation models of users and items have often relied on the idea of extrapolating preferences from similar users. Different machine learning models define the notion of similarity differently. Classical bi-linear Matrix Factorization (MF) approaches model users and items via their identifiers and represent them as vectors in the latent space [13; 28]. Modern deep-learning-based recommender systems [34; 12; 25], which are also used for predicting top-$k$ items given an item, learn the user-to-item propensities from large amounts of training data containing many (user, item) tuples, optionally with available item content information (e.g., product descriptions) and user metadata.

As machine learning models, the performance of RecSys relies heavily on the amount of training data available. This might be feasible for large e-commerce or content delivery websites such as Overstock and Netflix, but poses a serious chicken-and-egg problem for small startups, whose amount of data, in turn, relies on the performance of their RecSys. On the other hand, zero-shot learning promises some degree of generalization from an old dataset to an entirely new dataset. In this paper, we explore the possibility of zero-shot learning in RecSys. We develop an algorithm, dubbed ZEro-Shot Recommenders (ZESREC), that is trained on an old dataset and generalize to a new one where there are *neither overlapping users nor overlapping items*, a setting that contrasts typical cross-domain

RecSys that has either overlapping users or items [40; 38; 3; 16]. Naturally, generalization of RecSys to unseen users and unseen items becomes the two major challenges for developing zero-shot RecSys.

For the first challenge on unseen users, we build on a rich body of literature on sequential recommendation models [12; 27; 25; 14; 17; 21]. These models are built with sequential structure to encode temporal ordering of items in user's item interaction history. Representing users by the items they have consumed in the past allows the model to extrapolate the preference learning to even novel users who the model did not see during training, as long as the items these unseen users have interacted with have been seen during training. However, such deep learning models encode item via its categorical item index, i.e., the item ID, and therefore fall short in predicting a likely relevant but brand-new item not previously seen during training.

This brings us to the second challenge of developing zero-shot recommender systems, i.e., dealing with unseen items. To address this challenge, ZESREC goes beyond traditional categorical item indices and uses items' generic features such as natural-language descriptions, product images, and videos as their continuous indices, thereby naturally generalizing to any unseen items. Take natural-language (NL) descriptions as an example. One can think of NL descriptions as a system of universal identifiers that indexes items from arbitrary domains. Therefore as long as one model is trained on a dataset with NL descriptions, it can generalize to a completely different dataset with a similar NL vocabulary. In ZESREC we build on state-of-the-art pretrained NL embedding models such as BERT [8] to extract NL embeddings from raw NL descriptions, leading to an item ID system in the continuous space that is generalizable across arbitrary domains. For instance, in e-commerce platform, one could use items' description text; and similarly in the online content platforms, one could use movie synopsis or music track descriptions to represent an item.

Combining the merits of sequential RecSys and the idea of universal continuous ID space, our ZESREC successfully enables recommendation in an extreme cold-start setting, i.e., the zero-shot setting where all users and items in the target domain are unseen during training. Essentially ZESREC tries to learn transferable user behavioral patterns in a universal continuous embedding space. For example, in the source domain, ZESREC can learn that if users purchase snacks or drinks (e.g., 'Vita Coconut Water' with a lemonade flavor) that they like, they may purchase similar snacks or drinks with different flavors (e.g., 'Vita Coconut Water' with a pineapple flavor), as shown in the case study. Later in the target domain, if one user purchase 'V8 Splash' with a tropical flavor, ZESREC can recommend 'V8 Splash' with a berry flavor to the user (see Fig. 4 of Sec. 4.6 for details). Such generalization is possible due to the use of the NL descriptions as universal identifiers, based on which ZESREC could easily identify similar products of the same brand with different flavors. To summarize our contributions:

- We identify the problem of zero-shot recommender systems and propose ZESREC as the first hierarchical Bayesian model for addressing this problem.
- We introduce the notion of universal continuous identifiers that makes recommendation in a zero-shot setting possible.
- We provide empirical results which show that ZESREC can successfully recommend items in the zero-shot setting.
- We conduct case studies demonstrating that ZESREC can learn interpretable user behavioral patterns that can generalize across datasets.

## 2 RELATED WORK

**Deep Learning for RecSys.** Deep learning has been prevalent in modern recommender systems [29; 33; 34; 19; 4; 10; 32] due to its scalability and superior performance. As a pioneer work, [29] uses restricted Boltzmann machine (RBM) to perform collaborative filtering in recommender systems, however the system is a single-layer RBM. Later, [34] and [19] build upon Bayesian deep learning to develop hierarchical Bayesian models that tightly integrate content information and user-item rating information, thereby significantly improving recommendation performance. After that, there are also various proposed sequential (or session-based) recommender systems [12; 27; 2; 18; 22; 39; 14; 31; 41; 25], GRU4Rec [12] was first proposed to use gated recurrent units (GRU) [6] for recommender systems. Since then, follow-up works such as hierarchical GRU [27], temporal convolutional networks (TCN) [2], and hierarchical RNN (HRNN) [25] have achieved improvement in terms of accuracy by utilizing cross-session information [27], causal convolutions [2], as well as meta data and control

signals [25]. Another line of work focusing on building self-attention based sequential models such as SASRec [14], BERT4Rec [31], and S3Rec [41]. In this paper we build on such sequential RecSys and note that our ZESREC is model agnostic, that is, it is compatible with any sequential RecSys.

**Cross-Domain and Cold-Start RecSys.** There is a rich literature on cross-domain RecSys focusing on training a recommender system in the source domain and deploying it in the target domain where there exist either common users or items [40; 38; 3; 16]. These works are also related to the problem of recommendation for cold-start users and items, i.e., users and items with few interactions (or ratings) available during training [11; 20; 42; 23]. There are also works [24; 9] handling cold start on both user and item with meta-learning, however they cannot generalize across domains. In summary, prior systems are either (1) not sufficient to address our zero-shot setting where there are neither common users nor common items in the target domain or (2) unable to learn user behavior patterns that are transferable across datasets/domains. Therefore, they are not applicable to our problem of zero-shot recommendations.

## 3 ZERO-SHOT RECOMMENDER SYSTEMS

In this section we introduce our ZESREC which is compatible with any sequential model. Without loss of generality, here we focus on NL descriptions as a possible instantiation of universal identifiers, but note that our method is general enough to use as identifiers other content information such as items' images and videos. We leave exploration for other potential modalities to future work.

**Notation.** We focus on the setting of zero-shot recommendation where there are *neither overlapping users nor overlapping items* between a source domain and a target domain. We assume a set $\mathcal{V}_s$ of $J_s$ items and a set $\mathcal{U}_s$ of $I_s$ users in the source domain, as well as a set $\mathcal{V}_t$ of $J_t$ items and a set $\mathcal{U}_t$ of $I_t$ users in the target domain. We let $I = I_s + I_t$ and $J = J_s + J_t$; we use $j \in \mathcal{V}_s \cup \mathcal{V}_t$ to index items and $i \in \mathcal{U}_s \cup \mathcal{U}_t$ to index users. The zero-shot setting dictates that $\mathcal{V}_s \cap \mathcal{V}_t = \emptyset$ and that $\mathcal{U}_s \cap \mathcal{U}_t = \emptyset$. We denote the collection of all users' interactions as a 3D tensor (with necessary zero-padding) $\mathbf{R} \in \mathbb{R}^{I \times N_{max} \times J}$, where $N_{max}$ is the maximum number of interactions among all users. We use the subscript '$*$' to represent the collection of all elements in a certain dimension. Specifically, each user $i$ has a sequence of $N_i$ interactions (e.g., purchase history) with various items denoted as $\mathbf{R}_{i**} = [\mathbf{R}_{it*}]_{t=1}^{N_i}$, where $\mathbf{R}_{it*} \in \{0,1\}^J$ is one-hot vector denoting the $t$-th item user $i$ interacted with. The same user $i$ has different user embeddings at different time $t$, reflecting dynamics in user interests; here we denote as $\mathbf{u}_{it} \in \mathbb{R}^D$ the latent user vector when user $i$ interacts with the $t$-th item in her history, and we use $\mathbf{U} = [\mathbf{u}_{it}]_{i=1,t=1}^{I,N_{max}} \in \mathbb{R}^{I \times N_{max} \times D}$ (with necessary zero-padding) to denote the collection of user latent vectors. We denote as $\mathbf{v}_j \in \mathbb{R}^D$ the item latent vector and $\mathbf{V} = [\mathbf{v}_j]_{j=1}^J \mathbb{R}^{J \times D}$ as the collection. For simplicity and without loss of generality, in this paper we focus on using NL descriptions (i.e., a sequence of words) to describe items. For item $j$ we denote its NL description as $\mathbf{x}_j$, and the number of words as $M_j$; similar to $\mathbf{V}$, we let $\mathbf{X} = [\mathbf{x}_j]_{j=1}^J$. With slight notation overload on $t$, we denote as $\mathbf{R}^{(s)}$ and $\mathbf{R}^{(t)}$ the sub-tensor of $\mathbf{R}$ that corresponds to the source and target domains, respectively. Similarly, we also have $\mathbf{U}^{(s)}, \mathbf{U}^{(t)}, \mathbf{V}^{(s)}, \mathbf{V}^{(t)}, \mathbf{X}^{(s)}$, and $\mathbf{X}^{(t)}$.

**Problem Setup.** A model is trained using all users' interaction sequences from the source domain, i.e., $\{\mathbf{R}_{i**}\}_{i \in \mathcal{U}_s}$, and then deployed to recommend items for any user $\iota \in \mathcal{U}_t$ in the target domain, given user $\iota$'s previous history $\mathbf{R}_{\iota**}$, which can be empty. In practice we append a dummy item at the beginning of each user session, so that during inference we could conduct recommend even for users without any history by ingesting the dummy item as context to infer the user latent vector. In our zero-shot setting, the model is not allowed to fine-tune or retrain on any data from the target domain.

**Definition of Zero-shot Learning in RecSys.** Our zero-shot setting includes three unique properties: (1) cold users, (2) cold items, and (3) domain gap. It is fundamentally different from previous content-based cold start as the latter setting usually satisfies either (1) or (2), but not often both.

### 3.1 FROM CATEGORICAL DOMAIN-SPECIFIC ITEM ID TO CONTINUOUS UNIVERSAL ITEM ID

Most models in recommender systems learn item embeddings through interactions. These embeddings are indexed by *categorical domain-specific item ID*, which is transductive and cannot be generalized to unseen items.

In this paper, we propose to use item generic content information such as NL descriptions and image to produce item embeddings, which can be used as *continuous universal item ID*. Since such content information is domain agnostic, the model trained on top of it can be transferable from one domain to another, therefore making zero-shot recommender systems feasible. Based on the universal item embeddings, one can then build sequential models to obtain user embeddings by aggregating embeddings of items in the user history.

Here we introduce the notion of universal embedding networks (UEN) that use continuous universal embeddings to index items (and therefore users) rather than categorical ID that is not transferable across domains. We call the UEN generating item and user universal embeddings item UEN and user UEN, respectively.

## 3.2 MODEL OVERVIEW

We propose a hierarchical Bayesian model with a probabilistic encoder-decoder architecture. The encoder ingests items from user history to yield the user embedding, while decoder computes recommendation scores based on similarity between user embeddings and item embeddings.

**Generative Process.** The generative process of ZESREC (in the source domain) is as follows:

1. For each item $j$:
   - Compute the item universal embedding: $\mathbf{m}_j = f_e(\mathbf{x}_j)$.
   - Draw a latent item offset vector $\boldsymbol{\epsilon}_j \sim \mathcal{N}\left(\mathbf{0}, \lambda_v^{-1}\mathbf{I}_D\right)$.
   - Obtain the item latent vector: $\mathbf{v}_j = \boldsymbol{\epsilon}_j + \mathbf{m}_j$.
2. For each user $i$:
   - For each time step $t$:
     - Compute the user universal embedding: $\mathbf{n}_{it} = f_{seq}([\mathbf{v}_{i_\tau}]_{\tau=1}^{t-1})$.
     - Draw a latent user offset vector $\boldsymbol{\xi}_{it} \sim \mathcal{N}\left(\mathbf{0}, \lambda_u^{-1}\mathbf{I}_D\right)$.
     - Obtain the latent user vector: $\mathbf{u}_{it} = \boldsymbol{\xi}_{it} + \mathbf{n}_{it}$.
     - Compute recommendation score $\mathbf{S}_{itj}$ for each user-interaction-item tuple $(i, k, j)$, $\mathbf{S}_{itj} = f_{softmax}(\mathbf{u}_{it}^\top \mathbf{v}_j)$ and draw the $t$-th item for user $i$: $\mathbf{R}_{it*} \sim Cat([\mathbf{S}_{itj}]_{j=1}^J)$.

Here $f_{softmax}(\cdot)$ is the softmax function: $f_{softmax}(\mathbf{u}_{it}^\top \mathbf{v}_j) = \exp(\mathbf{u}_{it}^\top \mathbf{v}_j)/\sum_j \exp(\mathbf{u}_{it}^\top \mathbf{v}_j)$. $Cat(\cdot)$ is a categorical distribution. $f_e(\cdot)$ is item UEN (see Sec. 3.3), $f_{seq}(\cdot)$ is user UEN (see Sec. 3.4). $i_\tau$ in $\mathbf{v}_{i_\tau}$ indexes the $\tau$-th item that user $i$ interacts with. $\lambda_u$ and $\lambda_v$ are hyperparameters. The latent item offset $\boldsymbol{\epsilon}_j = \mathbf{v}_j - \mathbf{m}_j$ provides the final latent item vector $\mathbf{v}_j$ with the flexibility to slightly deviate from the content-based item universal embedding $\mathbf{m}_j$. Similarly, the latent user offset $\boldsymbol{\xi}_{it} = \mathbf{u}_{it} - \mathbf{n}_{it}$ provides the final latent user vector $\mathbf{u}_{it}$ with the flexibility to slightly deviate from the user universal embedding $\mathbf{n}_{it}$. Intuitively, $\boldsymbol{\epsilon}_j$ and $\boldsymbol{\xi}_{it}$ provide domain-specific information on top of the

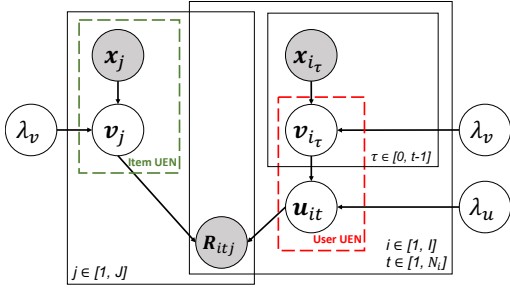

Figure 1: Graphical model for ZESREC. The item side (left) and the user side (right) share the same $\lambda_v$ and $\mathbf{v}$'s. The plates indicate replication.

domain-agnostic information from $\mathbf{m}_j$ and $\mathbf{n}_{it}$. In the target domain we will remove $\boldsymbol{\epsilon}_j$ from $\mathbf{v}_j$, and $\boldsymbol{\xi}_{it}$ from $\mathbf{u}_{it}$, which can be seen as an attempt to remove the bias learned from the source domain.

**Training.** The MAP estimation in the source domain can be decomposed as following:

$$P(\mathbf{U}^{(s)}, \mathbf{V}^{(s)}|\mathbf{R}^{(s)}, \mathbf{X}^{(s)}, \lambda_u^{-1}, \lambda_v^{-1}) \propto P(\mathbf{R}^{(s)}|\mathbf{U}^{(s)}, \mathbf{V}^{(s)}) \cdot P(\mathbf{U}^{(s)}|\mathbf{V}^{(s)}, \lambda_u^{-1}) \cdot P(\mathbf{V}^{(s)}|\mathbf{X}^{(s)}, \lambda_v^{-1}).$$

Maximizing the posterior probability is equivalent to minimizing the joint Negative Log-Likelihood (NLL) of $\mathbf{U}^{(s)}$ and $\mathbf{V}^{(s)}$ given $\mathbf{R}^{(s)}$, $\mathbf{X}^{(s)}$, $\lambda_u^{-1}$, and $\lambda_v^{-1}$:

$$\mathcal{L} = \sum_{i=1}^{I_s} \sum_{t=1}^{N_i} -\log(f_{softmax}(\mathbf{u}_{it}^\top \mathbf{v}_{it})) + \frac{\lambda_u}{2} \sum_{i=1}^{I_s} \sum_{t=1}^{N_i} ||\mathbf{u}_{it} - f_{seq}(\{\mathbf{v}_{i_\tau}\}_{\tau=1}^{t-1})||_2^2 + \frac{\lambda_v}{2} \sum_{i=1}^{J_s} ||\mathbf{v}_j - f_e(\mathbf{x}_j)||_2^2, \quad (1)$$

where $\mathbf{n}_{it} = f_{seq}(\{\mathbf{v}_{i_\tau}\}_{\tau=1}^{t-1})$ and $\mathbf{m}_j = f_e(\mathbf{x}_j)$. See the Appendix for a full Bayesian treatment of ZESREC using inference networks and generative networks [15].

**Inference and Recommendation in the Target Domain.** Once the model is trained using source-domain data, it can recommend unseen items $j \in \mathcal{V}_t$ (where $\mathcal{V}_t \cap \mathcal{V}_s = \emptyset$) for any unseen user $i \in \mathcal{U}_t$ (where $\mathcal{U}_t \cap \mathcal{U}_s = \emptyset$) from the target domain based on the approximate MAP inference below:

$$p(\mathbf{R}^{(t)}|\mathbf{X}^{(t)}) = \int p(\mathbf{R}^{(t)}|\mathbf{U}^{(t)}, \mathbf{V}^{(t)}, \mathbf{X}^{(t)})p(\mathbf{U}^{(t)}, \mathbf{V}^{(t)}|\mathbf{X}^{(t)})d\mathbf{U}^{(t)}d\mathbf{V}^{(t)}$$

$$\approx \int p(\mathbf{R}^{(t)}|\mathbf{U}^{(t)}, \mathbf{V}^{(t)}, \mathbf{X}^{(t)})\delta_{\mathbf{U}_{MAP}^{(t)}}(\mathbf{U}^{(t)})\delta_{\mathbf{V}_{MAP}^{(t)}}(\mathbf{V}^{(t)})d\mathbf{U}^{(t)}d\mathbf{V}^{(t)},$$

where $\delta(\cdot)$ denotes a Dirac delta distribution. $\mathbf{U}_{MAP}^{(t)}$ and $\mathbf{V}_{MAP}^{(t)}$ are the MAP estimate of $\mathbf{U}^{(t)}$ and $\mathbf{V}^{(t)}$ given $\mathbf{X}^{(t)}$, which we approximate as:

$$(\mathbf{U}_{MAP}^{(t)}, \mathbf{V}_{MAP}^{(t)}) \approx \underset{\mathbf{U}^{(t)}, \mathbf{V}^{(t)}}{\operatorname{argmax}} \, p(\mathbf{U}^{(t)}, \mathbf{V}^{(t)}|\mathbf{X}^{(t)}) = \left( f_{seq}(f_e(\mathbf{X}^{(t)})), f_e(\mathbf{X}^{(t)}) \right). \quad (2)$$

The reason for the approximation is that ZESREC has no access to interactions $\mathbf{R}^{(t)}$ in the target domain, making the cross-entropy loss in the Eqn. 4 disappear. The user and item latent matrices $\mathbf{U}_{MAP}^{(t)}, \mathbf{V}_{MAP}^{(t)}$ in the target domain enable us to perform zero-shot recommendation by computing recommendation scores based on inner products and recommend item $\operatorname{argmax}_j f_{softmax}(\mathbf{u}_{it}^\top \mathbf{v}_j)$.

As long as the unseen items' NL descriptions are available, ZESREC could obtain both the unseen users' latent vectors and the unseen items' latent vectors based on the item universal embedding network. This is in contrast to previous methods that rely on catogorical domain-specific item ID, which is not transferable as unseen items have completely different ID from items in the training set. Note that ZESREC is general enough to adapt to other data modalities such as images and videos.

### 3.3 ITEM UNIVERSAL EMBEDDING NETWORK: $\mathbf{m}_j = f_e(\mathbf{x}_j)$

The purpose of the item universal embedding network, denoted as $f_e(\cdot)$, is to extract item embeddings that are universal across domains. The network consists of a pretrained BERT network [8], denoted as $f_{BERT}$, followed by a single-layer neural network, denoted by $f_{NN}(\cdot)$. Formally we have

$$\mathbf{m}_j = f_e(\mathbf{x}_j) = f_{NN}(f_{BERT}(\mathbf{x}_j)), \quad (3)$$

where $\mathbf{m}_j$ is the universal embedding for item $j$ and $\mathbf{x}_j$ is the NL description for item $j$. We use the embedding for the 'CLS' token from the last layer of BERT as the output of $f_{BERT}(\mathbf{x}_i)$. This UEN is jointly trained with the sequential model using the objective function in Eqn. 4. Note that $f_{NN}(\cdot)$ is necessary as we need to adapt the pre-trained BERT for recommendation tasks.

### 3.4 USER UNIVERSAL EMBEDDING NETWORK: $\mathbf{n}_{it} = f_{seq}([\mathbf{v}_{i_\tau}]_{\tau=1}^{t-1})$

The user UEN $f_{seq}(\cdot)$ is built on top of the item UEN in Eqn. 3. Specifically, given user $i$'s interaction sequence until time $t$, $\mathbf{R}_{i**} = [\mathbf{R}_{i\tau*}]_{\tau=1}^{t-1}$, we first replace it the corresponding NL descriptions $\mathbf{l}_i^{(x)} = [\mathbf{x}_{i_\tau}]_{\tau=1}^{t-1}$, and then fed it into item UEN in Eqn. 3 to obtain $\mathbf{l}_i^{(m)} = [\mathbf{m}_{i_\tau}]_{\tau=1}^{t-1} = [f_e(\mathbf{x}_{i_\tau})]_{\tau=1}^{t-1}$ where item $i_t$ is user $i$'s $t$-th interaction. Each vector in the sequence, $\mathbf{m}_j$, is the universal embedding for item $j$. During training, we obtain item latent vector $\mathbf{v}_j$ based on $\mathbf{m}_j$ (see Eqn. 4), while during inference $\mathbf{v}_j = \mathbf{m}_j$ (see Eqn. 2). We can then treat each $\mathbf{v}_j$ as the input at each time step of the sequential model, which gives us the final user universal embedding $\mathbf{n}_{it} = f_{seq}([\mathbf{v}_{i_\tau}]_{\tau=1}^{t-1})$. Note that this user UEN is used during both training (Eqn. 4) and inference (Eqn. 2).

## 4 EXPERIMENTS

In this section, we evaluate our ZESREC against various in-domain and zero-shot baselines on three source-target dataset pairs, with the major goals of addressing the following questions:

**Q1** How accurate (effective) is ZESREC compared to the baselines?

**Q2** If one allows training models using target-domain data, how long does it take for non zero-shot models to outperform zero-shot recommenders?

**Q3** Does ZESREC yield meaningful recommendations for users with similar behavioral patterns in the source domain and target domain?

Table 1: Zero-shot results on three dataset pairs, 'Amazon Grocery and Gourmet Food' → 'Amazon Prime Pantry', 'Others' → 'NFL', and 'Others' → 'NCAA'. Methods such as HRNN, HRNN-Meta, and POP are *oracle methods* that are trained directly using target-domain data. N@20 and R@20 represent NDCG@20 and Recall20, respectively. The top 3 zero-shot results are shown in bold.

| Method | Prime Pantry | | NCAA 1st day | | NCAA 1 week | | NFL 1st day | | NFL 1 week | |
|---|---|---|---|---|---|---|---|---|---|---|
| | N@20 | R@20 | N@20 | R@20 | N@20 | R@20 | N@20 | R@20 | N@20 | R@20 |
| HRNN (Oracle) | 0.038 | 0.073 | 0.066 | 0.139 | 0.006 | 0.011 | 0.052 | 0.118 | 0.002 | 0.003 |
| HRNN-Meta (Oracle) | 0.045 | 0.089 | 0.054 | 0.120 | 0.004 | 0.010 | 0.044 | 0.112 | 0.001 | 0.003 |
| GRU4Rec (Oracle) | 0.042 | 0.081 | 0.062 | 0.135 | 0.005 | 0.011 | 0.046 | 0.109 | 0.001 | 0.003 |
| GRU4Rec-Meta (Oracle) | 0.044 | 0.088 | 0.053 | 0.118 | 0.004 | 0.009 | 0.037 | 0.094 | 0.001 | 0.003 |
| TCN (Oracle) | 0.038 | 0.073 | 0.068 | 0.141 | 0.006 | 0.012 | 0.049 | 0.114 | 0.001 | 0.003 |
| TCN-Meta (Oracle) | 0.045 | 0.088 | 0.054 | 0.120 | 0.004 | 0.010 | 0.044 | 0.109 | 0.001 | 0.003 |
| POP (Oracle) | 0.007 | 0.018 | 0.002 | 0.005 | 0.000 | 0.000 | 0.000 | 0.000 | 0.000 | 0.000 |
| EMB-KNN (Baseline) | 0.024 | 0.042 | 0.016 | 0.026 | 0.005 | 0.011 | **0.010** | **0.018** | 0.001 | 0.003 |
| Random (Baseline) | 0.001 | 0.002 | 0.002 | 0.006 | 0.002 | 0.006 | 0.001 | 0.002 | 0.001 | 0.002 |
| ZESRec-H (Ours) | **0.027** | **0.052** | **0.027** | **0.063** | **0.011** | **0.036** | 0.007 | 0.013 | **0.015** | **0.043** |
| ZESRec-G (Ours) | **0.026** | **0.050** | **0.030** | **0.070** | **0.014** | **0.040** | **0.008** | 0.013 | **0.018** | **0.058** |
| ZESRec-T (Ours) | **0.026** | **0.050** | **0.023** | **0.056** | **0.011** | **0.035** | **0.009** | **0.017** | **0.018** | **0.054** |

## 4.1 Datasets

We use three different real-world dataset pairs, one from Amazon [26] and two from MIND [37]:

- **Amazon** [26]: A publicly available dataset collection which contains a group of datasets in different categories with abundant item metadata such as item description, product images, etc. In our experiments, we consider two datasets: (1) 'Prime Pantry', which contains 300K interactions, 10K items, and 76K users, and (2) 'Grocery and Gourmet Food', which contains 2.3M interactions, 213K items, and 739K users.
- **MIND** [37]: A large-scale news recommendation dataset collected from the user click logs of Microsoft News. It includes 4-week user history and 5th-week interactions. In our experiments, we simulate zero-shot learning by leave-one-out splitting on subcategories under the largest category in MIND: 'sports'. We consider two pairs: (1) 'Others' to 'NFL', where 'Others' contains all the other subcategories except 'NFL'. 'Others' contains 169K interactions, 8K items, and 57K users, while 'NFL' contains 1M interactions, 11K items, and 203K users. (2) 'Others' to 'NCAA', where 'Others' includes all the subcategories except 'NCAA'. 'Others' contains 797K interacions, 17K items, and 206K users, while 'NCAA' contains 238K interactions, 31K items, and 56K users.

We adopted a rigorous experimental setup for zero-shot learning to ensure (1) **no overlapping** users and items and (2) **no temporal leakage**. See the Appendix for details. Datasets in Amazon pair are divided into training (80%) and test (20%) sets. For datasets in the two MIND pairs, we follow the official splitting [37] and use the four-week user history as the training set and interactions in the fifth week as the test set. Due to strong recency bias in the news recommendation [37], for MIND we run experiments under two settings to measure temporal invariance of the model: (1) evaluate on interactions in the *first day* of the week (**1st-day setting**), and (2) evaluate on interactions of the *whole week*, i.e., the full test set (**whole-week setting**). For all datasets, We further split 10% of the training set by user as validation set.

## 4.2 Baselines and ZESRec Variants

To demonstrate the effectiveness of our model, we compare ZESRec against two groups of baselines: in-domain methods and zero-shot methods.

**In-Domain Methods.** We compare variants of our model ZESRec against a variety of state-of-the-art session-based recommendation models including **GRU4Rec** [12], **TCN** [2], and **HRNN** [25]. We also consider their extensions, **HRNN-Meta**, **GRU4Rec-Meta**, and **TCN-Meta**, which use items' NL description embeddings to replace item ID hidden embeddings. Besides aforementioned sophisticated models, we introduce **POP** which is a simple baseline conducting recommendation only based on item global popularity. It can be a strong baseline in certain domains. All the above 7 methods are trained directly on target-domain data and therefore are considered *'oracle'* methods.

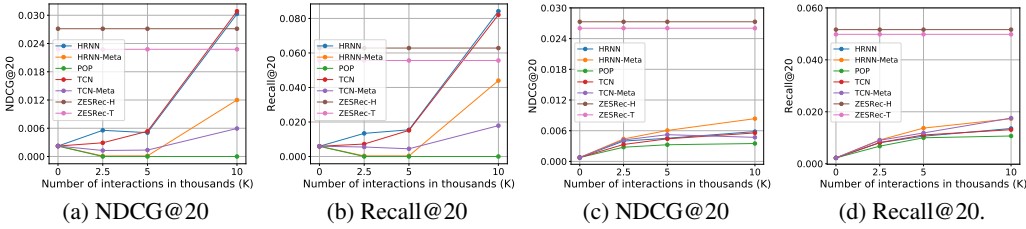

| (a) NDCG@20 | (b) Recall@20 | (c) NDCG@20 | (d) Recall@20. |

Figure 2: Incremental training results for baselines using target domain data compared to ZESREC using *no training data* on MIND-NCAA (left two) and Amazon Prime Pantry (right two). To prevent clutter, we only show results for TCN-based and HRNN-based models, since HRNN is an advanced version of GRU4Rec. Results show that even without using target-domain data, ZESREC can still outperform models trained directly using target-domain data for substantial amount of time.

**Zero-Shot Methods.** Since no previous work has been done on this thread, we consider two intuitive zero-shot models (1) **EmbeddingKNN**: a K-nearest-neighbors algorithm based on inner product between the user embedding and item embedding. The item embedding is the BERT embedding from NL description, while the user embedding is an average over embeddings of the user's interacted items, and (2) **Random**: recommending items by random selection from the whole item catalogue without replacement.

**ZESREC Variants.** We evaluate three variants of our ZESREC, including **ZESREC-G**, **ZESREC-T**, and **ZESREC-H** which use GRU4Rec, TCN and HRNN as base models, respectively.

### 4.3    IMPLEMENTATION DETAILS

We adopted pre-trained *google/bert_uncased_L-12_H-768_A-12* BERT model from Huggingface [36] to process item description and generate item embedding. The dimension of BERT embedding is 768. For the model architecture, we use BERT embedding as input to a single-layer neural netwrok (i.e., $f_{NN}(\cdot)$). The output dimension for the NN is set to $D$ (see the Appendix for more details).

### 4.4    EXPERIMENT SETUP AND EVALUATION METRICS

We conducted three sets of experiments to answer questions proposed earlier in this section.

**Zero-Shot Experiments.** We trained in-domain baselines on target domain training set, while our ZESREC is trained on source domain. All models are tested on the testing set of the target domain for an apples to apples comparison.

**Incremental Training Experiments.** To measure how long it takes for non-zero-shot models to outperform zero-shot recommenders, we conducted incremental training experiments on in-domain base models GRU4Rec, TCN, HRNN as well as GRU4Rec-Meta, TCN-Meta, HRNN-Meta. Note that the variants of our ZESREC are NOT retrained on target domain. It is also *inevitable* that non-zero-shot models eventually outperform ZESREC because ZESREC does not have access to target-domain data. Due to space constraint, we only reported results for incremental training experiments on Amazon pair and MIND NCAA pair under the 1st-day setting. Importantly, under the whole-week setting for the MIND dataset, the ZESRec variants already significantly outperform in-domain competitors which are trained on the full target training set; therefore it is meaningless to compare incremental training results under this setting, as in-domain competitors will never beat ZESRec variants. For all the source-target dataset pairs, we group the interactions by user and sort interactions within each user based on interaction timestamps. We randomly select users until we get enough interactions and build three datasets containing 2.5K, 5K, and 10K interactions, respectively.

**Case Studies.** To gain more insight what ZESREC learns, we perform several case studies. Specifically, we randomly select users from the test set of the target domain Amazon Prime Pantry (where we evaluate ZESREC) and only keep users for whom ZESREC correctly predicts the 6-th items in the sequence given the first 5 items as context, as we want to focus on sequences where ZESREC works. We use these users as queries to find users with similar behavioral patterns from the source domain Amazon Grocery and Gourmet Food (where we train ZESREC) based on user embeddings from ZESREC. User embeddings are generated based on the first 5 items of the sequence.

**Simulated Online Scenarios.** In the experiments, ZESRec only accesses target domain data during inference to simulate online scenarios, where new businesses just open and the customers are using

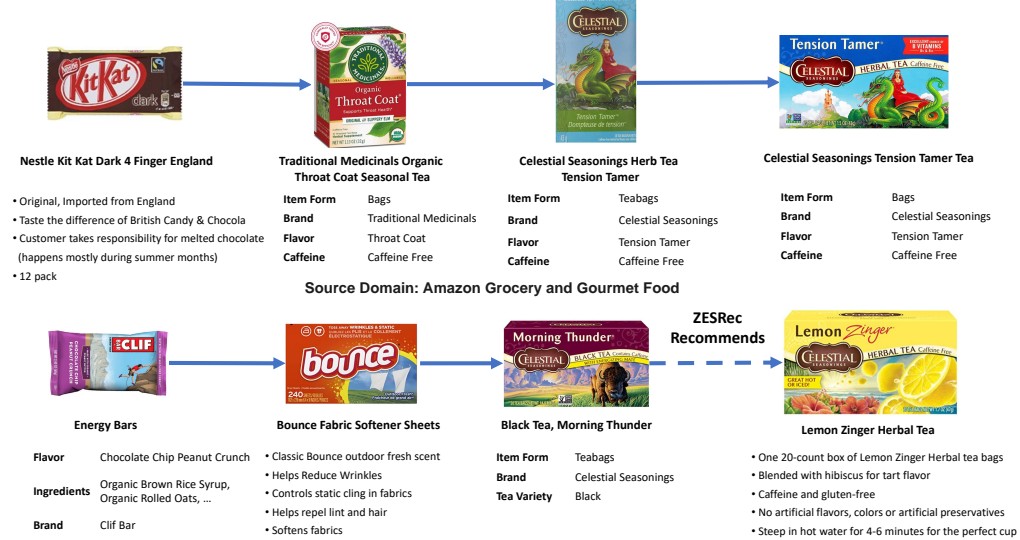

Figure 3: Case Study 1. The purchase history of a user in the source domain (*top*) and the purchase history of an unseen user in the target domain, where all items are unseen during training (*bottom*). We select two users with similar universal embeddings according to Sec. 4.4. This case study demonstrates ZESREC can learn the user behavioral pattern that 'users who bought sugary snacks and tea tend to buy caffeine-free herbal tea later'.

service in real-time. This real-time online access setting is substantially different from batch access ahead of time as it prevents us from training a recommender in the target domain before serving.

**Evaluation Protocol.** For evaluation, we adopted Recall (R@20) and the ranking metric Normalized Discounted Cumulative Gain (NDCG) [30] (N@20). We removed all the repetitive interactions (e.g., user A clicked item B two times in a row) to only focus on evaluating the model's capability of capturing the transition between user history to the next item.

## 4.5 EXPERIMENTAL RESULTS

**Zero-Shot Experiments.** The experimental results on three dataset pairs are in Table 1. Overall, our ZESREC outperforms zero-shot baselines Embedding-KNN and Random by a large margin in most cases; it can also achieve performance comparable to in-domain baselines. On the Amazon pair, ZESREC beats POP and achieves comparable performance with HRNN-Meta and HRNN-Interactions, suggesting the existence of shared recommendation patterns in two domains. For the two MIND pairs, we made the following observations: (1) Under the whole-week setting, ZESREC consistently outperforms HRNN-Meta and HRNN-Interactions by a large margin. Comparing the first-day results and the whole-week results, it is obvious that the in-domain models overfit the history and fail to generalize well on latest user-item interactions. This reflects strong recency bias in news recommendation. In contrast, our zero-shot learning naturally comes with an inductive bias to only model the transition from user history to the next item. (2) Under the first-day setting, ZESREC still achieves reasonable performance comparing with in-domain models on both pairs. (3) Surprisingly in the MIND NFL pair, the ZESRec variants perform well under the whole-week setting even when the source domain contains much fewer interactions than the target domain (169K VS 1M).

**Incremental Training Experiments.** Incremental training results are in Fig. 2. Overall, almost all the in-domain baselines are not able to outperform ZESREC by retraining on at most 10K interactions in target domain, which shows the vast importance of conducting zero-shot learning in recommender domain. For new business operating an early-stage RecSys, it's hard to train a good RecSys with limited interactions. This is a chicken-and-egg problem, as training good RecSys requires sufficient interactions, while in turn, collecting sufficient interactions requires a satisfactory RecSys to attract users. Therefore the first 10K interactions are crucial to get the RecSys started.

## 4.6 CASE STUDIES

To gain more insight, we randomly select pairs of user purchase histories from the source and target domains in the Amazon pair as case studies according to the procedure in Sec. 4.4. The goal is

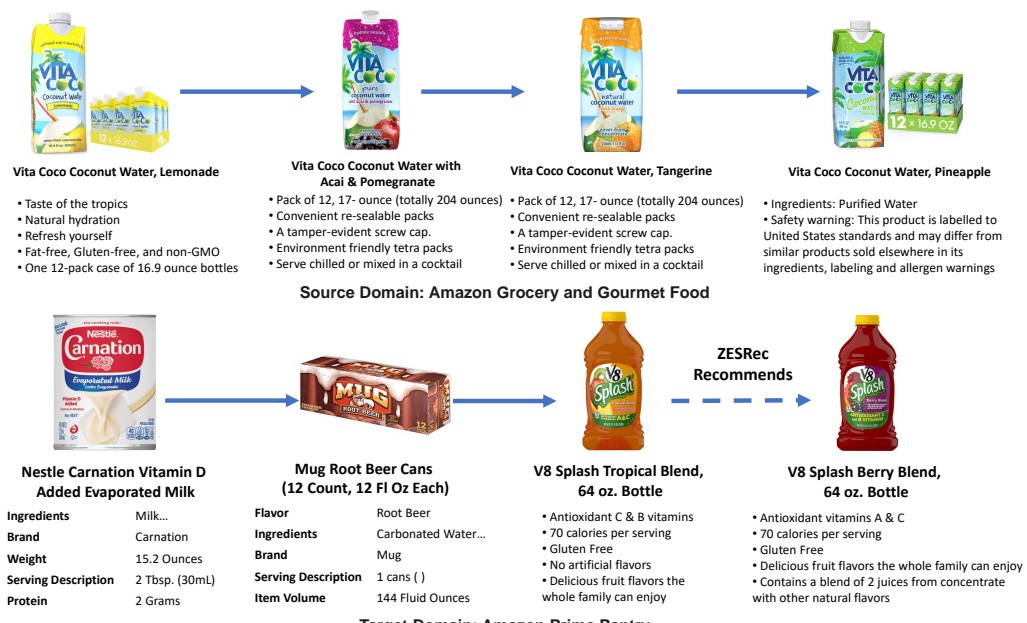

Figure 4: Case Study 2. The purchase history of a user in the source domain (*top*) and the purchase history of an unseen user in the target domain, where all items are unseen during training (*bottom*). We select two users with similar universal embeddings according to Sec. 4.4. This case study demonstrates ZESREC can learn the user behavioral pattern that 'if users bought snacks or drinks that they like, they may later purchase similar snacks or drinks with different flavors'.

to demonstrate our ZSR could learn relevant dynamics of users' purchase history from the source domain and successfully recommend unseen products to an unseen user in the target domain.

Fig. 3 shows the purchase history of a user in the source domain (*top*), 'Amazon Grocery and Gourmet Food', and the purchase history of an unseen user in the target domain (*bottom*), 'Amazon Prime Pantry', where all items are unseen during training. The user in the source domain bought 'Tension Tamer Tea', which is a type of herbal tea, after buying some sugary snacks (KitKat) and other tea. Such a pattern is captured by ZESREC, which then recommended 'Lemon Zinger Herbal Tea' to an unseen user after she bought some sugary snacks ('Energy Bars from Clif Bar') and some black tea. This case study demonstrates ZESREC can learn the user behavioral pattern that 'users who bought sugary snacks and tea tend to buy caffeine-free herbal tea later'. More interestingly, another case study in Fig. 4 demonstrates that ZESREC can learn the user behavioral pattern that 'if users bought snacks or drinks that they like, they may later purchase similar snacks or drinks with different flavors'. Specifically, in the source domain, the user purchased 'Vita Coconut Water' with four different flavors; such a pattern is captured by ZESREC. Later in the target domain, an unseen user purchase 'V8 Splash' with a tropical flavor, ZESREC then successfully recommends 'V8 Splash' with a berry flavor to the user.

## 5 CONCLUSION

In this paper, we identify the problem of zero-shot recommender systems where a model is trained in the source domain and deployed in a target domain where all the users and items are unseen during training. We propose ZESREC as the first general framework for addressing this problem. We introduce the notion of universal continuous identifiers leveraging the fact that item ID can be grounded in natural-language descriptions. We provide empirical results, both quantitatively and qualitatively, to demonstrate the effectiveness of our proposed ZESREC and verify that ZESREC can successfully learn user behavioral patterns that generalize across datasets (domains). The main limitation of this work is that commonality between domains is crucial for ZESRec, as is the case for most other works in transfer learning. Future work includes quantification of such commonality and exploring other modalities, e.g., images and videos, as alternative universal identifiers. It would also be interesting to investigate the interpretability provided by the pretrained BERT model and to incorporate additional auxiliary information to further improve the zero-shot performance.

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

# A APPENDIX

## A.1 DATA USAGE AND PRIVACY DISCUSSIONS

**Data Usage:** For real-world scenarios, new business owners who hope to do zero-shot recommendation need to check the dataset policy before usage. On the other hand, if multinational corporations hope to establish a branch in a new region they could use their own data from other existing regions.

**Privacy:** To protect the privacy of user data, we encourage people who want to adopt our methods to train the model using source-domain data with differential privacy embedded. Some related references include [1; 5; 35].

## A.2 BAYESIAN TREATMENT WITH INFERENCE NETWORKS

**Training.** Using Jensen's inequality we have the following evidence lower bound (ELBO):

$$\log p(\mathbf{R}|\mathbf{X}, \lambda_u, \lambda_v) \geq E_q[\log p(\mathbf{R}, \mathbf{U}, \mathbf{V}|\mathbf{X}, \lambda_u, \lambda_v)] - E_q[\log q(\mathbf{U}, \mathbf{V}|\mathbf{X})],$$

where the expectation is over $q(\mathbf{U}, \mathbf{V}|\mathbf{X})$. We have

$$\mathcal{L} = \sum_{i=1}^{I_s} \sum_{t=1}^{N_i} -\log(f_{softmax}(\mathbf{u}_{it}^\top \mathbf{v}_{i_t})) + \frac{\lambda_u}{2} \sum_{i=1}^{I_s} \sum_{t=1}^{N_i} ||\mathbf{u}_{it} - f_{seq}(\{\mathbf{v}_{i_\tau}\}_{\tau=1}^{t-1})||_2^2 + \frac{\lambda_v}{2} \sum_{j=1}^{J_s} ||\mathbf{v}_j - f_e(\mathbf{x}_j)||_2^2,$$

(4)

For the variational distribution $q(\mathbf{U}, \mathbf{V}|\mathbf{X})$ we have the following factorization:

$$q(\mathbf{U}, \mathbf{V}|\mathbf{X}) = q(\mathbf{V}|\mathbf{X})q(\mathbf{U}|\mathbf{V}, \mathbf{X}) = q(\mathbf{V}|\mathbf{X})q(\mathbf{U}|\mathbf{V}),$$

where $\mathbf{U} = [\mathbf{u}_i]_{i=1}^I$, $\mathbf{V} = [\mathbf{v}_j]_{j=1}^J$, and $\mathbf{X} = [\mathbf{x}_j]_{j=1}^J$

$$q(\mathbf{v}_j|\mathbf{X}) = q(\mathbf{v}_j|\mathbf{x}_j) = \mathcal{N}\Big(\mathbf{v}_j; f_{e,\mu}(\mathbf{x}_j), f_{e,\sigma^2}(\mathbf{x}_j)\Big),$$

$$q(\mathbf{u}_i|\mathbf{V}, \mathbf{X}) = \mathcal{N}\Big(\mathbf{u}_i; f_{seq,\mu}(\{\mathbf{v}_{j_k}\}_{k=1}^{n_i}), f_{seq,\sigma^2}(\{\mathbf{v}_{j_k}\}_{k=1}^{n_i})\Big)$$

Overall,

$$\log p(\mathbf{R}|\mathbf{X}, \lambda_u, \lambda_v) \geq E_q[\log p(\mathbf{R}, \mathbf{U}, \mathbf{V}|\mathbf{X}, \lambda_u, \lambda_v)] - E_q[\log q(\mathbf{U}, \mathbf{V}|\mathbf{X})]$$

$$= E_q[\sum_{i=1}^{I_s} \sum_{k=1}^{N_i} \log(f_{softmax}(\mathbf{u}_{ik}^T \mathbf{v}_{i_k}))]$$

$$+ E_q[\log p(\mathbf{V}|\mathbf{X})] + E_q[\log p(\mathbf{U}|\mathbf{V})] - E_q[\log q(\mathbf{U}, \mathbf{V}|\mathbf{X})] + C$$

$$= E_q[\sum_{i=1}^{I_s} \sum_{k=1}^{N_i} \log(f_{softmax}(\mathbf{u}_{ik}^T \mathbf{v}_{i_k}))] - \frac{\lambda_v}{2} \sum_j \|f_{e,\sigma^2}(\mathbf{x}_j)\|_1$$

$$+ E_q[\log p(\mathbf{U}|\mathbf{V})] - E_q[\log q(\mathbf{U}, \mathbf{V}|\mathbf{X})] + C,$$

where the expectation is over $q(\mathbf{U}, \mathbf{V}|\mathbf{X})$. Below we discuss several important terms including $-E_q[\log q(\mathbf{U}, \mathbf{V}|\mathbf{X})]$ and $E_{q(\mathbf{U},\mathbf{V}|\mathbf{X})}[\log p(\mathbf{U}|\mathbf{V})]$ in the ELBO.

The term $-E_q[\log q(\mathbf{U}, \mathbf{V}|\mathbf{X})]$ is the entropy of $q(\mathbf{U}, \mathbf{V}|\mathbf{X})$.

$$-E_{q(\mathbf{U},\mathbf{V}|\mathbf{X})}[\log q(\mathbf{U}, \mathbf{V}|\mathbf{X})] = -E_{q(\mathbf{V}|\mathbf{X})} E_{q(\mathbf{U}|\mathbf{V})}[\log q(\mathbf{V}|\mathbf{X}) + \log q(\mathbf{U}|\mathbf{V})]$$

$$= -E_{q(\mathbf{V}|\mathbf{X})} E_{q(\mathbf{U}|\mathbf{V})}[\log q(\mathbf{V}|\mathbf{X})] - E_{q(\mathbf{V}|\mathbf{X})} E_{q(\mathbf{U}|\mathbf{V})}[\log q(\mathbf{U}|\mathbf{V})]$$

$$= -E_{q(\mathbf{V}|\mathbf{X})}[\log q(\mathbf{V}|\mathbf{X})] - E_{q(\mathbf{V}|\mathbf{X})}\Big[E_{q(\mathbf{U}|\mathbf{V})}[\log q(\mathbf{U}|\mathbf{V})]\Big]$$

$$= \frac{1}{2} \sum_j [\mathbf{1}^\top \log f_{e,\sigma^2}(\mathbf{x}_j)] + \frac{1}{2} \sum_i E_{q(\mathbf{V}|\mathbf{X})}\Big[\sum_{t=1}^{N_i} \mathbf{1}^\top \log f_{seq,\sigma^2}(\{\mathbf{v}_{i_\tau}\}_{\tau=1}^{t-1})\Big] + C$$

$$\approx \frac{1}{2} \sum_j [\mathbf{1}^\top \log f_{e,\sigma^2}(\mathbf{x}_j)] + \frac{1}{2N_v} \sum_i \sum_{\mathbf{V}} \Big[\sum_{t=1}^{N_i} \mathbf{1}^\top \log f_{seq,\sigma^2}(\{\mathbf{v}_{i_\tau}\}_{\tau=1}^{t-1})\Big] + C,$$

where $\mathbf{V}$ is sampled for $N_v$ times to get a Monte Carlo estimate of $E_{q(\mathbf{V}|\mathbf{X})}[\log f_{seq,\sigma^2}(\mathbf{v}_{i_\tau}\}_{\tau=1}^{t-1})]$. In practice, it is found that one sample is usually sufficient due to the use of SGD-based optimization process.

Similarly the term

$$E_{q(\mathbf{U},\mathbf{V}|\mathbf{X})}[\log p(\mathbf{U}|\mathbf{V})] = E_{q(\mathbf{V}|\mathbf{X})}\Big[E_{q(\mathbf{U}|\mathbf{V})}[\log p(\mathbf{U}|\mathbf{V})]\Big]$$

$$= -\frac{\lambda_u}{2} E_{q(\mathbf{V}|\mathbf{X})}\Big[\sum_{i=1}^{I_s} \sum_{t=1}^{N_i} \|f_{seq,\sigma^2}(\{\mathbf{v}_{i_\tau}\}_{\tau=1}^{t=1})\|_1\Big] + C$$

$$= -\frac{\lambda_u}{2N_v} \sum_{\mathbf{V}} \sum_{i=1}^{I_s} \sum_{t=1}^{N_i} \|f_{seq,\sigma^2}(\{\mathbf{v}_{i_\tau}\}_{\tau=1}^{t-1})\|_1 + C,$$

where $\mathbf{V}$ is sampled for $N_v$ times from $q(\mathbf{V}|\mathbf{X})$ to get a Monte Carlo estimate of $E_{q(\mathbf{V}|\mathbf{X})}[\|f_{seq,\sigma^2}(\mathbf{v}_{i_k}\}_{k=1}^{n_i})\|_2^2]$. In practice, it is found that one sample is usually sufficient due to the use of SGD-based optimization process.

**Inference.** Inference can be done via Monte Carlo estimates of $p(\mathbf{R}|\mathbf{X}, \lambda_u, \lambda_v)$. Specifically,

$$p(\mathbf{R}|\mathbf{X}, \lambda_u, \lambda_v) = E_q(p(\mathbf{R}|\mathbf{U}, \mathbf{V})) \approx \frac{1}{N_v N_u} \sum_{\mathbf{V}^{(n)}} \sum_{\mathbf{U}^{(n)}} p(\mathbf{R}|\mathbf{U}^{(n)}, \mathbf{V}^{(n)}),$$

where $\mathbf{V}^{(n)} \sim q(\mathbf{V}|\mathbf{X})$ and $\mathbf{U}^{(n)} \sim q(\mathbf{U}|\mathbf{V}^{(n)}, \mathbf{X})$.

One could also use MAP inference to trade accuracy for speed.

$$p(\mathbf{R}|\mathbf{X}) = \int p(\mathbf{R}|\mathbf{U}, \mathbf{V}, \mathbf{X}) p(\mathbf{U}, \mathbf{V}|\mathbf{X}) d\mathbf{U} d\mathbf{V} \approx \int p(\mathbf{R}|\mathbf{U}, \mathbf{V}, \mathbf{X}) \delta_{\mathbf{U}_{MAP}}(\mathbf{U}) \delta_{\mathbf{V}_{MAP}}(\mathbf{V}) d\mathbf{U} d\mathbf{V},$$

where $\delta(\cdot)$ denotes a Dirac delta distribution. $\mathbf{U}_{MAP}$ and $\mathbf{V}_{MAP}$ are the MAP estimate of $\mathbf{U}$ and $\mathbf{V}$ given $\mathbf{X}$:

$$(\mathbf{U}_{MAP}, \mathbf{V}_{MAP}) \approx \underset{\mathbf{U},\mathbf{V}}{\arg\max} \, q(\mathbf{U}, \mathbf{V}|\mathbf{X}) = \Big(f_{seq}(f_{e,\mu}(\mathbf{X})), f_{e,\mu}(\mathbf{X})\Big).$$

### A.3 IMPLEMENTATION DETAILS

We use pre-trained *google/bert_uncased_L-12_H-768_A-12* BERT model from Huggingface [36] to process item description and generate item embedding. The dimension of BERT embedding is 768. We use BERT embedding as input to a single-layer neural netwrok (NN) and the output dimension for the NN is set to $D$, which equals to the hidden dimension of the sequential model.

For ZESREC variants we use the default optimal setting: we set the hidden dimension $D$ as 300, the dropout rate as 0.2, and the number of training epochs as 20. We use Adagrad [7] as the optimizer with a learning rate of 0.1, and train ZESREC variants in the source domain with early stopping based on validation loss. We set the hyperparameter $\lambda_v$ as a relatively large value 100 to restrain the variance of the item offset vector $\epsilon_j$.

For base models (HRNN, TCN, GRU4Rec) and corresponding base-meta models (HRNN-Meta, TCN-Meta, GRU4Rec-Meta), we set the dropout rate as 0.2 and the number of training epochs as 20; we choose Adagrad [7] as the optimizer. We train base models and base-meta models in the target domain with early stopping and perform hyperparameter tuning, both are based on the validation loss. We tried the hidden dimension $D$ in $\{128, 300\}$ and the learning rate $\eta$ in $\{0.01, 0.1, 1\}$, and choose to use the configurations $\{D : 128, \eta : 1\}$ for base models and $\{D : 128, \eta : 0.1\}$ for base-meta models.

For all datasets, we treat the rating as implicit feedback (interactions between user and item). Since we are considering session-based recommendation and using sequential model, we filter out users with only 1 interaction as the sequential model need to ingest at least one item from user history as context to perform next-step prediction.

**No Temporal Leakage.** We adopted a rigorous experimental setup for zero-shot learning: Firstly, we makes sure that there are **no** overlapping users and items between the source domain and the target domain. Secondly, we temporally split the two domains, meaning that all the training interactions in the source domain must be happened before all the testing interactions in the target domain.

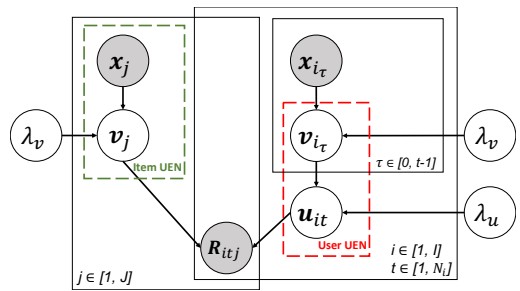

Figure 5: Graphical model for ZESREC. The item side (left) and the user side (right) share the same $\lambda_v$ and **v**'s. The plates indicate replication.

All experiments were ran on a GPU machine with Nvidia Tesla V100 16G memory GPU.

### A.4 MODEL ARCHITECTURE

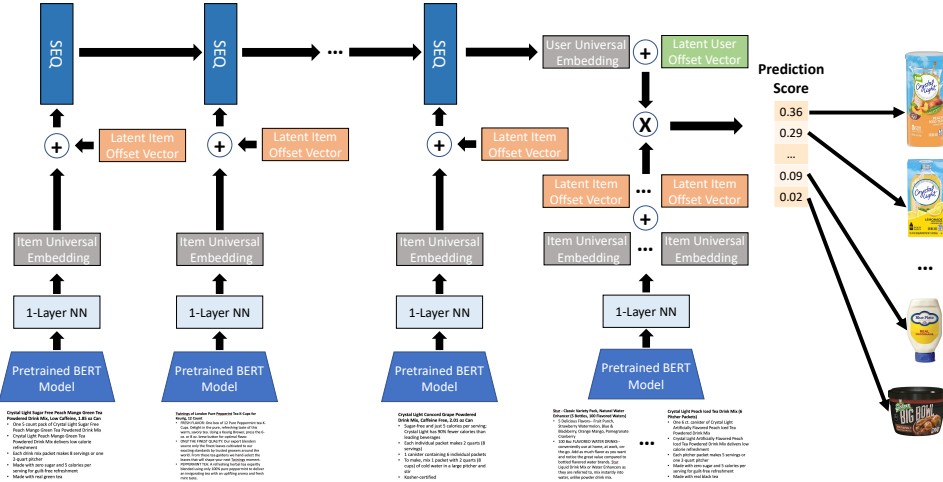

Figure 6: Model Architecture

The graphical model for ZESREC is shown in Fig. 5. In the MAP estimation version of ZES-REC, we set $\lambda_u \to \infty$ to remove latent user offset vector, thereby preventing ZESREC from over-parameterization. Fig. 6 shows a simplified deterministic model architecture from the neural network point of view. Below we elaborate on the process in terms of two stages: training in source domain and inference in target domain.

**Training in Source Domain.** During training, on the encoder side, we generate BERT embeddings from items' NL descriptions as item universal embeddings and then add the learnable item offset vectors to them, which yield the final item embeddings (item latent vectors). The sequential model will aggregate item embeddings of items in user history to generate user embeddings. On the decoder side, we compute item latent vectors the same way we do on the encoder side. Here we share item latent vectors on both encoder and decoder sides to reduce number of parameters. Empirically we find this prevent overfitting and improve performance.

**Inference in Target Domain.** In this phase, we will remove the item offset vectors on both the encoder and decoder sides. We use item universal embeddings directly instead as the final item latent vectors since we have no access to interactions in the target domain, and therefore the model is unable to estimate the item offset vectors.

