# OpenReview forum: "Zero-Shot Recommender Systems"
_ICLR.cc/2022/Conference — ICLR 2022 Submitted_

### Official Review · Reviewer_hJB8 · 2021-11-01

**Correctness:** 3
**Technical Novelty And Significance:** 1
**Empirical Novelty And Significance:** 2
**Recommendation:** 3
**Confidence:** 4

**Main Review:**

This paper studies "zero shot recommendation", where there is no overlap between the source domain and target domain in terms of user id and item ids. This is different from the recent cross-domain recommendations where overlapping items are leveraged. The main idea is to use item content features that can be generalizable, such as using BERT over item descriptions. Users are represented based on the items that were interacted by them. Experiments are conducted on two offline datasets, where variants of the proposed methods outperform some in-domain only methods and simple cross-domain baselines (such as random and direct embedding matching). Some case studies are provided.

Strength
- The paper is clearly written and the authors clearly communicates what has been done

Weakness
- The major issue of this paper is novelty. The reviewer agrees that the "zero shot recommendation" is probably an interesting future direction, but the way that this problem is tackled in this paper makes it not different from known problem settings. More specifically, the paper uses content-based methods instead of ids. Content-based recommendation is as old as id based recommendations, if not older. Content-based recommendations is the default solution for cold-start problems and is commonly used together with ids in practice. The method proposed in this paper is not different from existing content-based recommendation paradigm:
1. The model is learned on source domain and directly applied to target domain. There are no tuning on target domain (e.g., by using some unsupervised methods). The important assumption here is, the source and target domain are actually still *in-domain*, otherwise the model will just fail since nothing is done in the target domain. Then it is not really different from content-based methods for cold start problem, which has been studied for decades. The reviewer understands for real-world applications that are different problems, but in terms of machine learning (the focus of this conference), the reviewer does not see novelty. Actually, in terms of applications:
2. One motivation to do "zero shot recommendations" is that small new websites do not have enough data. However, the paper assumes that users in the target domain has interactions available (otherwise there will be no user representation). This is confusing since the reviewer does not understand if this is really zero-shot or not. Many recommendation papers use such datasets so it is a standard data setting. Furthermore, the experiments are conducted on two offline datasets and the authors need to do a bunch of data massaging to make it "zero shot". This makes the reviewer wonder how valuable this setting is - if user interactions are already available in the target domain, why "zero shot" is needed? Also, ideally there should be real-world use cases to show this setting can really benefit new websites.

Overall, the reviewer recommends the authors to think carefully what "zero shot recommendation" is a novel meaningful setting and why the proposed methods in this paper has any meaningful novelties. Currently the problem setting and proposed method do not seem to really show novelty, regardless of the bar of ICLR.

Reply to rebuttal:
The reviewer acknowledge the response but not convinced.

1. The authors argue that "Our zero-shot setting is fundamentally different from content-based cold start".

- The reviewer understands the difference of the setting and mentioned in the original review that this setting can have values (if tackled concretely). The reviewer's point is, the proposed method makes assumptions that make the "new setting" work only under scenarios that is virtually the same as cold start or cross-domain recommendation. More specifically, only inference is done on the target domain without any adjustment. This will clearly fail in many cases. The authors keep using claims such as "completely different", "extreme cold-start" - but what would happen, for example, if the model learned on MIND dataset is applied to Amazon datasets? How would the owner of a new website decide which dataset to use? How would they get such datasets?

2. The authors argue that "Our setting is zero-shot because the target domain data is not used during training, but only during inference to simulate online scenarios, where new businesses just open and the customers are using service in real-time. What the reviewer refers to is batch access ahead of time, which is distinct from our online access case." The authors also argue that "Previous initial-phase recommenders can only collect low-quality interactions" and "Initial-phase recommenders take much longer time to collect data because inaccurate recommendations are not appealing to users." The authors referred to Figure 2, the "incremental training", to argue for effectiveness.
- The reviewer finds some arguments and the incremental training experiments confusing. First, data used to train recommender systems are not necessarily from recommendation UI. It is common practice to just use "interaction" data, regardless of where it comes from. In fact, one can argue that interactions not from the UI do not possess many kinds of biases and are of higher quality. Second, Figure 2 is not really measuring what the authors are arguing about. Noticeably, the proposed methods have constant performance over the time period (while other methods get better as more training data is available). How could the proposed method work well when there are no interactions (as its inference depends on user's past interactions)? The reviewer notices that the test set is on *later date* (i.e., week 5 for MIND) and the proposed method *already leverages previous interactions* in first 4 weeks during inference. Again, I understand it is not trained to update any parameters, but think what would happen in practice - such interaction data is available anyway (for the proposed model to do inference). a) The performance in day1 or week1 is not measured, which is what the authors are really arguing about. On the real day1 without any interactions, the proposed method will generate nothing (or the same set of items to all users). Does it really help bootstrapping a new website? 2) If the proposed method has first 4 weeks data for inference, then other methods should use them for training - it is just the proposed method can't train with them. So the reviewer feels the experiments are not fair under the current setting. Please let me know if I misunderstood anything in terms of the experimental setting.




**Summary Of The Paper:**

This paper studies "zero shot recommendation" where source and target domain have no overlap in terms of user and items. The paper proposes to use item content features, such as leveraging BERT on descriptions, instead of IDs. Experiments are conducted on two offline datasets.

**Summary Of The Review:**

Though the paper is well written, the reviewer finds it difficult to believe the "zero shot recommendation" is a novel setting and the proposed methods have differ from content traditional content-based recommender systems in meaningful ways.

---

> ### Author Response · Authors · 2021-11-14
> **Author Response for Reviewer hJB8 Part 2**
>
> **Q3.1 One motivation to do "zero shot recommendations" is that small new websites do not have enough data. However, the paper assumes that users in the target domain has interactions available (otherwise there will be no user representation). This is confusing since the reviewer does not understand if this is really zero-shot or not. ... - if user interactions are already available in the target domain, why "zero shot" is needed?**
>
> We do not assume the interactions are already available in the target domain, please note that:
>
> **1. Simulated online adoption process: the target domain data is not available for training.** Our setting is zero-shot because the target domain data is not used during training, but only during inference to simulate online scenarios, where new businesses just open and the customers are using service in real-time. What the reviewer refers to is *batch access ahead of time*, which is distinct from our *online access case*.
>
> **2. ZESRec is valuable even if target domain data is collected online for offline training.**
>
> **(1)** **Not enough data:** Available interactions in the target domain are not sufficient to train a good in-domain model during the early stages of the product/service. As shown in Figure 2 of the paper, even after 10K interactions, the trained in-domain model still underperforms our proposed zero-shot model ZESRec. In reality, it may take a few weeks or months to collect data of such volume.
>
> **(2)** **Differences in feedback loop: data quality and collection speed:** Our incremental training experiment is offline thus highly in favor of the in-domain models regarding data quality and collection speed:
>
>      **(i) Data quality:** Good recommenders like ZESRec could collect high-quality data. In contrast, previous initial-phase recommenders can only collect low-quality interactions, thus they need many more interactions to achieve the same performance as ZESRec.
>
>      **(ii) Collection speed:**  Initial-phase recommenders take much longer time to collect data because inaccurate recommendations are not appealing to users.
>
> **Q3.2 Also, ideally there should be real-world use cases to show this setting can really benefit new websites.**
>
> We observe demands along this direction - companies hoping to launch services in new regions. However, to provide such services, we must collect sufficient experimental results to gain confidence in our models. The results from this paper can serve as a starting point.

---

> ### Author Response · Authors · 2021-11-14
> **Author Response for Reviewer hJB8 Part 1**
>
> **Q1 The major issue of this paper is novelty. ... the way that this problem is tackled in this paper makes it not different from known problem settings. ... The method proposed in this paper is not different from existing content-based recommendation paradigm**
>
> Our ZESRec is substantially different from classical content-based methods:
>
> **(1)** **Novel Bayesian Modeling:** We proposed a novel hierarchical Bayesian model (Figure 1) for neural sequential recommenders. It is methodologically novel even for in-domain models as it is different from previous non-probabilistic sequential recommenders. Preliminary results show it is competitive to most recent sequential recommenders.
>
> **(2)** **Domain Debiasing:** We proposed a novel debiasing technique. Specifically: (i) when training ZESRec in source domains: we use the latent item bias vector as a way of capturing the bias introduced by the training data; (ii) when making predictions in target domains: we remove the latent item bias vector (see Section 3.2 *Inference and recommendation in the target domain* for the derivation).
>
> **(3)** **A Full Bayesian Treatment to Handle Uncertainty and Potentially Further Improve Performance:** With our formulation, the model can be easily extended to a fully Bayesian model with potentially even better performance (see the Appendix for details).
>
> **Q2 ... The important assumption here is, the source and target domain are actually still in-domain, ... Then it is not really different from content-based methods for cold start problem, ... in terms of machine learning (the focus of this conference), the reviewer does not see novelty. Actually, in terms of applications**
>
> **1. Definition of Zero-shot Learning in Recommendation:** Our zero-shot setting is fundamentally different from content-based cold start, with three unique properties: (1) cold users, (2) cold items, and (3) domain gap. Content-based usually satisfies either (1) or (2), but not often both. To handle these three properties, we propose a hierarchical Bayesian method for debiasing (Property 3) while coping with cold users (Property 1) and cold items (Property 2) simultaneously.
>
> In line with the aforementioned definition, in our experiments we ensure that for all dataset pairs there are 0 overlapping users or items between source domains and target domains. More importantly, the items in source domains and target domains are from *different categories*.
>
> **2. Other Differences with Content-based Methods:** ZESRec is significantly different from the previous content-based methods in two aspects:
>
> **(1)** We proposed the first hierarchical Bayesian deep learning model for this new zero-shot problem. It considers domain debiasing and is novel even for in-domain recommenders. Please also refer to our response to Q1.
>
> **(2)** Previous works have a more rigorous requirement on item attribute overlapping, meaning that both domains need to share the exact same set of item attributes (e.g., movie genres). In contrast, ZESRec only loosely requires *general unstructured NL description without a prescribed format*, which is *available in virtually all domains*.

---

> ### Author Response · Authors · 2021-11-20
> **A Gentle Reminder**
>
> A gentle reminder: Does our response address your concerns? We look forward to your feedback; any suggestions/criticisms would be highly appreciated.

---

### Official Review · Reviewer_zQRZ · 2021-11-02

**Correctness:** 4
**Technical Novelty And Significance:** 2
**Empirical Novelty And Significance:** 3
**Recommendation:** 5
**Confidence:** 4

**Main Review:**

Strengths:

1 The authors identify a new and interesting problem called zero-shot recommendation.

2 The proposed zero-shot recommendation method is simple and generic.

Weakness:

1 The idea is a bit too straightforward, i.e., using the attributes of the items/users and their embeddings to bridge any two domains.

2 The technical contribution is limited, i.e., there is no significant technical contribution and extension based on a typical model for the cross-domain recommendation setting.



**Summary Of The Paper:**

In this paper, the authors identify a new and interesting problem called zero-shot recommendation, where there is no overlap of users or items between a source domain and a target domain. The main idea is to bridge two domains via the attributes of the item and the users, which is called continuous universal item ID and user ID.

**Summary Of The Review:**

In this paper, the authors identify a new and interesting problem called zero-shot recommendation, where there is no overlap of user or items between a source domain and a target domain. The main idea is to bridge two domains via the attributes of the item/users, which is called continuous universal item/user ID.

The authors conduct experiments and show that the proposed zero-shot recommendation framework work well.

Overall, the paper is well presented. The studied problem (zero-shot recommendation) is interesting.

However, my main concern is that the idea to bridge two non-overlap domains via attributes is too straightforward, and the technical contribution is limited.

---

> ### Author Response · Authors · 2021-11-14
> **Author Response for Reviewer zQRZ**
>
> **Q1 The idea is a bit too straightforward, i.e., using the attributes of the items/users and their embeddings to bridge any two domains.**
>
> We argue that:
>
> **(1)** We are the first to identify the zero-shot problem in recommender systems which is an undiscovered essential task with huge potential business impact.
>
> **(2)** We proposed the first hierarchical Bayesian deep learning model for this new problem. For details, please refer to our response to Q2.
>
> **(3)** Why do we need complexity if we could effectively solve this essential problem with an elegant, interpretable, and not-so-complex solution? Besides, our framework is flexible; it can build upon and improve any complex sequential recommenders.
>
> **Q2 The technical contribution is limited, i.e., there is no significant technical contribution and extension based on a typical model for the cross-domain recommendation setting.**
>
> We argue that our technical contributions are:
>
> **(1)** **Novel Bayesian Modeling:** We proposed a novel hierarchical Bayesian model (Figure 1) for neural sequential recommenders. It is methodologically novel even for in-domain models as it is different from previous non-probabilistic sequential recommenders. Preliminary results show it is competitive to most recent sequential recommenders.
>
> **(2)** **Domain Debiasing:** We proposed a novel debiasing technique. Specifically: (i) when training ZESRec in source domains: we use the latent item bias vector as a way of capturing the bias introduced by the training data; (ii) when making predictions in target domains: we remove the latent item bias vector (see Section 3.2 *Inference and recommendation in the target domain* for the derivation).
>
> **(3)** **A Full Bayesian Treatment to Handle Uncertainty and Potentially Further Improve Performance:** With our formulation, the model can be easily extended to a fully Bayesian model with potentially even better performance (see the Appendix for details).

---

> ### Author Response · Authors · 2021-11-20
> **A Gentle Reminder**
>
> A gentle reminder: Does our response address your concerns? We look forward to your feedback; any suggestions/criticisms would be highly appreciated.

---

### Official Review · Reviewer_cG8S · 2021-11-02

**Correctness:** 4
**Technical Novelty And Significance:** 3
**Empirical Novelty And Significance:** 3
**Recommendation:** 6
**Confidence:** 4

**Details Of Ethics Concerns:**

Since this paper is about using some dataset to train and generalize it to other datasets, are there any concern in terms of legal aspects as well as compliance of whether you can use some datasets for other tasks?


**Main Review:**

Pros:
1. This paper is well motivated. The problem of zero-shot recommendation is interesting yet very challenging to solve.
2. The proposed approach with universal item embedding combined with sequential recommendation methods for user embedding levering the items that user have interacted with is a novel and reasonable approach for solving this problem.
3. The paper demonstrate the effectiveness of the proposed approach using real-world dataset and experiments. The questions in Section 4 that the experiment tries to answer are informative. The experiment set up is solid with following both the non-overlapping as well as temporal aspect to it.
4. It is good that the paper demonstrates the proposed framework using multiple base sequential models.

Cons:
1. The literature survey does not cover a lot of more recent approaches from sequential recommendations. Although the proposed approach is model agnostic, it would still be good a provide a thorough literature review. Same for the baselines compared are not as strong. Some examples papers (there are more) are below:
Sun, Fei, et al. "BERT4Rec: Sequential recommendation with bidirectional encoder representations from transformer." Proceedings of the 28th ACM international conference on information and knowledge management. 2019.
Kang, Wang-Cheng, and Julian McAuley. "Self-attentive sequential recommendation." 2018 IEEE International Conference on Data Mining (ICDM). IEEE, 2018.
Zhou, Kun, et al. "S3-rec: Self-supervised learning for sequential recommendation with mutual information maximization." Proceedings of the 29th ACM International Conference on Information & Knowledge Management. 2020.
2. It would be good to add some discussions in terms of how easy/hard the proposed approach can generalize to other modalities such as video, image, attributes. Does these modality also share the commonalities that language use?
3. What is the sequential model used for 3.4 user universal embedding network after getting each time steps embedding for each item?
4. In figure 2, when saying the ZSESRec do not directly use the target-domain data, do you use the interaction histories of users. If so, it will be consider to be as using the target data. If not, how is user represented, aren’t all uses will get the same results if no item is recommended?
5. The limitations in terms how close the source and target domain needs to be are not discussed in the paper.



**Summary Of The Paper:**

A great recommender system relies on great training set. However, at the beginning, there is no such data availability. This paper tries to solve the zero-shot recommendation problem where there is no user or item overlaps. The two challenges are generalize to unseen users and to unseen items. For unseens users, sequential recommendation represents users as a sequence of their interacted items. As long as the items are seen before, the users can be represented. As for items, the unique ids are not useful. However, the attributes such as natural language description can be universal. This paper proposed an approach based on hierarchical Bayesian model. The item universal embedding is using pre-trained BERT network with a single layer neural network. Extensive experiments are carried out to demonstrate the effectiveness of the proposed approach.


**Summary Of The Review:**

Overall, the problem being studied is interesting and challenging. The approach proposed in this paper make sense. The experiments questions and design are good. Many major concern is intermittent question 4 in Main Review. Please clarify more for the authors. There are some areas for improvement like baselines, literature review, lack of discussion about .

---

> ### Author Response · Authors · 2021-11-14
> **Author Response for Reviewer cG8S Part 2**
>
> **Q6 Details Of Ethics Concerns: Since this paper is about using some dataset to train and generalize it to other datasets, are there any concern in terms of legal aspects as well as compliance of whether you can use some datasets for other tasks?**
>
> This is a great point! To answer this question:
>
> **(1)** All the datasets in our paper are public and free for research usage. For real-world scenarios, new business owners who want to do zero-shot recommendation need to check the dataset policy. On the other hand, if the multinational corporations hope to establish a branch in a new region, they could simply use their own data from other existing regions.
>
> **(2)** Regarding privacy, one potential method is to train the model using source-domain data with differential privacy embedded. Some related references include [2, 3, 4].
> We have updated our Appendix with these points in the revision.
>
> **Reference**
>
> **[1]** Ma, Yifei, et al. "Temporal-Contextual Recommendation in Real-Time." Proceedings of the 26th ACM SIGKDD International Conference on Knowledge Discovery & Data Mining. 2020.
>
> **[2]** Abadi, Martin, et al. "Deep learning with differential privacy." Proceedings of the 2016 ACM SIGSAC conference on computer and communications security. 2016.
>
> **[3]** Chen, Mia Xu, et al. "Gmail smart compose: Real-time assisted writing." Proceedings of the 25th ACM SIGKDD International Conference on Knowledge Discovery & Data Mining. 2019.
>
> **[4]** Yu-Xiang Wang, et al. "Subsampled Renyi Differential Privacy and Analytical Moments Accountant.". in AISTATS-2019.

---

> ### Author Response · Authors · 2021-11-14
> **Author Response for Reviewer cG8S Part 1**
>
> **Q1 The literature survey does not cover a lot of more recent approaches from sequential recommendations. Although the proposed approach is model agnostic, it would still be good a provide a thorough literature review. Same for the baselines compared are not as strong. Some examples papers (there are more) are below: … BERT4Rec.. SASRec … S3-rec...**
>
> Thanks for your suggestions! We will cite and discuss the connections to these related work. In our experience, these models perform similarly on many datasets. Note that these models cannot handle content information like our ZESRec.
>
> We choose a representative set of models (including HRNN [1] published in 2020) to focus on our true contributions in the ZESRec setting. Our proposed approach can be augmented to the related architectures as well, which by default do not come equipped with our Bayesian interpretations.
>
> **Q2 It would be good to add some discussions in terms of how easy/hard the proposed approach can generalize to other modalities such as video, image, attributes. Does these modality also share the commonalities that language use?**
>
> This is a good point. We believe if other modalities such as images and videos are as informative to the recommendation task as text, we could simply replace the text BERT to other encoders such as image ResNet to do zero-shot recommendation. We also mentioned in the paper that our framework is generic and can be applied to content information other than text.
>
> In the paper we focus on text because there are not many recommendation datasets containing multi-modal data, especially images and videos. In general the textual data is easier to access than other types of content information.
>
> **Q3 What is the sequential model used for 3.4 user universal embedding network after getting each time steps embedding for each item?**
>
> The formulation in Section 3.4 is applicable to any base sequential models. Specifically, for HRNN and GRU4Rec we use the GRU to aggregate item embeddings of items in user history to generate user embeddings, while for TCN we use CNN to do such aggregation.
>
> **Q4 In figure 2, when saying the ZSESRec do not directly use the target-domain data, do you use the interaction histories of users. If so, it will be consider to be as using the target data. If not, how is user represented, aren’t all uses will get the same results if no item is recommended?**
>
> We are not using the target data during training but only during inference for online simulation, please note that:
>
> **1. Simulated online adoption process: the target domain data is not available for training.** Our setting is zero-shot because the target domain data is not used during training, but only during inference to simulate online scenarios, where new businesses just open and the customers are using service in real-time. What the reviewer refers to is *batch access ahead of time*, which is distinct from our *online access case*.
>
> **2. ZESRec is valuable even if target domain data is collected online for offline training.**
>
> **(1)** **Not enough data**: Available interactions in the target domain are not sufficient to train a good in-domain model during the early stages of the product/service. As shown in Figure 2 of the paper, even after 10K interactions, the trained in-domain model still underperforms our proposed zero-shot model ZESRec. In reality, it may take a few weeks or months to collect data of such volume.
>
> **(2)** **Differences in feedback loop: data quality and collection speed**: Our incremental training experiment is offline thus highly in favor of the in-domain models regarding data quality and collection speed:
>
>      **(i) Data quality:** Good recommenders like ZESRec could collect high-quality data. In contrast, previous initial-phase recommenders can only collect low-quality interactions, thus they need many more interactions to achieve the same performance as ZESRec.
>
>      **(ii) Collection speed:**  Initial-phase recommenders take much longer time to collect data because inaccurate recommendations are not appealing to users.
>
> **Q5 The limitations in terms how close the source and target domain needs to be are not discussed in the paper.**
>
> Thanks for the suggestion! We agree that commonality between domains is crucial for ZESRec, as is the case for most other works in transfer learning. We have updated the paper accordingly.

---

> ### Author Response · Authors · 2021-11-20
> **A Gentle Reminder**
>
> A gentle reminder: Does our response address your concerns? We look forward to your feedback; any suggestions/criticisms would be highly appreciated.

---

### Author Response · Authors · 2021-11-14
**General Response to All Reviewers**

We are highly grateful for all the constructive comments from reviewers. Here we summarize all the changes we made in the paper (highlighted in red) according to reviewers’ suggestions:

**(1)** Add definition of zero-shot RecSys in Section 3 (Reviewer hJB8).

**(2)** Add explanation of simulated online scenarios in Section 4.4 (Reviewer hJB8).

**(3)** Fix Figure 2 caption. Note that in simulated online scenarios, target domain data is only used during inference, preventing us from using it during training. (Reviewer cG8S).

**(4)** Add self-attention based recommenders in Section 2 (Reviewer cG8S).

**(5)** Add data usage and privacy discussions in Section A.1 (Reviewer cG8S).

**(6)** Further discuss limitations of our work in Conclusion (Reviewer cG8S).

---

### Author Response · Authors · 2021-12-02
**Looking forward to further discussion**

Dear reviewers, we are looking forward to further discussion on how to improve our paper, and we would be grateful for any suggestions/criticisms.

---

### Decision · Program_Chairs · 2022-01-20

**Decision:**

Reject

**Comment:**

The authors propose zero-shot recommendations, a scenario in which knowledge from a recommender system enables a second recommender system to provide recommendations in a new domain (i.e. new users & new items). The idea developed by the authors is to transfer knowledge through the item content information and the user behaviors.

The initial assessment of the reviewers indicated that this paper was likely not yet ready for publication. The reviewers all recognized the potential usefulness of zero-shot recommendations but argued that the implications of the proposed setup were somewhat unclear. Most notably, the reviewers raised the issue of how widely applicable this was in terms of distance between source and target domains (presumably the quality of the zero-shot recommendations depends on the distance).

The reviewers also noted that this was an application paper. This is of course within the CFP, and recommender systems papers have been published at ICLR in the past (for example one of the initial Session-based RecSys paper w. RNNs) but the potential audience for this work is somewhat lower at ICLR. I should also add that I agree with the authors that their model is novel, but it's very much tailored to this application and it was unclear to me how it might be impactful on its own. All in all, this did not play a significant role in my recommendation.

During the discussion, there were significant, yet respectful, disagreements between the authors and the reviewers. It also seems like perhaps the authors missed an important reply from reviewer hJB8 made available through their updated review (see "Reply to rebuttal"). So the discussion between reviewers and authors did not converge. Having said that, even the two most positive reviewers have scores that would make this paper a very borderline one (a 6 and a 5).

Further, I do find that reviewer's hJB8 arguments have merit and require another round of review. In particular, I think the role and effect of your simulated online scenario should be further discussed (note that I did read the new paragraph on it from your latest manuscript). For example, comparing to a baseline that can train with the data from this new domain would be useful even if at some point it ends up being an upper bound on the performance of your approach. I also found the question raised by the reviewer around the MIND results to be pertinent. Further characterizing pairs of domains in which the approach/works fails (even if empirically) would add depth to this paper.

All in all, this paper has interesting ideas and I strongly encourage the authors to provide a more thorough experimental setup that fully explores the benefits and limitations of their zero-shot approach.